# A Systematic Review of Construction and Demolition Waste Management in Australia: Current Practices and Challenges

Kamyar Kabirifar *, Mohammad Mojtahedi and Cynthia Changxin Wang

School of Built Environment, University of New South Wales, Sydney, NSW 2052, Australia; m.mojtahedi@unsw.edu.au (M.M.); cynthia.wang@unsw.edu.au (C.C.W.)
* Correspondence: kamyar.kabirifar@unsw.edu.au

**Abstract:** Construction and demolition waste (C&DW) has a deleterious impacts on sustainability not only in developing countries but also in developed nations. For example, Australia generated more than 27 million tonnes of C&DW in 2018–2019; however, only 60% of this waste stream was recovered. Considering this low recovery rate, lower than many developed nations, and with regards to the increasing rate of C&DW generation, extra attention should be given to the construction and demolition waste management (C&DWM) in Australia. Therefore, this research attempts to accurately understand the current practices and challenges of C&DWM in Australia. To do so, primarily, a systematic review of studies relevant to C&DWM from 2010 to 2021 was performed. In this step, 26 research documents were meticulously analysed to identify the current practices of C&DWM in Australia. Then, an in-depth interview with three experts were undertaken to verify the major results and to investigate the challenges of C&DWM in Australia. The results indicated that three factors significantly affect C&DWM in Australia, namely attitudes and behaviour of C&DWM stakeholders, C&DWM in project life cycles, and C&DWM regulations with regards to sustainability, adding that the latter was revealed as the most effective in C&DWM in Australia.

**Keywords:** construction and demolition waste management; systematic review; attitudes and behaviour of C&DWM stakeholders; C&DWM regulations with regards to sustainability; C&DWM in project life cycles; Australia

## 1. Introduction

The waste and resource recovery industry contributes more than $50 billion to the Australian economy per year (The Waste Management Association of Australia (WMRR)) [1], in which 56% of the waste-related activities are conducted by private and public trading waste management enterprises, 20% by the local governments, and 24% by the firms in other industries [2]. National Waste Policy (NWP) provides the solid waste laws and regulations in Australia [1]. Federal government, state and territory governments, and local governments (three Australian government levels) are engaged in waste management practices [1]. The federal government is responsible for providing national leadership, coordination between state and territory governments, and confirming the alignment of Australia's international constraints and obligations on waste management [1]. State and territory governments should manage the domestic waste. For instance, enforcing landfill levies, imposition rules for operating a landfill facility, license issuance for recycling facilities, license issuance for waste transportation, providing incentives for waste reuse and recycling, considering and assigning environmental protection measures with regard to illegal dumping or dump of hazardous waste [1], and local governments are the most involved organisations in general waste management practices such as collection, sorting, processing, and disposal [1,3] while focusing on sustainability development goals in waste management [3]. Whilst, for enhancing the waste diversion practice and releasing the landfills' pressure, waste management should obtain the sustainability movement not only from the government, but from the industry as well [4,5].

In Australia, waste is categorized into three main classes of Commercial and Industrial (C&I) waste, Construction and Demolition waste (C&DW), and Municipal Solid Waste (MSW) [6]. C&DW, which is the focus of this research, fundamentally includes masonry wastes such as asphalt, concrete, plasterboard, and bricks, organics (such as timber), metals as steel and aluminium, plastics, glass, paper and cardboard, textiles, leather, rubber, and others [3,6,7]. In 2018–2019, Australia generated more than 27 million tonnes (MT) of C&DW, accounting for roughly 44% of the total generated waste in Australia [6]. It should also be noted that approximately 60% of this waste was recycled or recovered; however, more than 36% was disposed [6,8]. The highest waste recycling rate is for South Australia (SA) with 80% and the lowest rate is for Northern Territory (NT) with 19%. Despite the C&DWM advancements in Australia, the output is far from optimum. For instance, the recycling rate of C&DW in Australia is still lower than some developed countries such as the Netherlands, which has an almost 90% recycling rate [9,10]. In addition, the increasing amount of C&DW generation in Australia (32% in the last thirteen years) [6], along with the detrimental impacts of C&DW on the environment, economy and the society (e.g., air pollution, environmental degradation, resource depletion, and global warming) [11–13], have prompted the need for further research into C&DWM in Australia. This research has been designed to address the following research question.

RQ: What are the current practices and challenges of C&DWM in Australia?

This research attempts to precisely understand the current practices and challenges of C&DWM in Australia. To do so, at first, a systematic review of relevant studies to C&DWM in Australia from 2010 to 2021 was performed to highlight the most updated practices. This approach extends the literature by a vast coverage in both time spanned topics and precise content analysis to reveal the current practices of C&DWM research in Australia. Then, an in-depth interview with three experts with extensive experience in C&DWM was undertaken to verify the results and investigate the challenges of C&DWM in Australia. The rest of this study is structured as follows: Section 2 explains the materials and methods of the study; Section 3 discusses the results extracted from current practices of C&DWM in Australia (from Section 2); Section 4 explains the in-depth interview procedure for results verification; and Section 5 concludes the study.

## 2. Materials and Methods

To reach the research objective, two research procedures have been performed. First, a systematic literature review was carried out to address the current practices of C&DWM in Australia. The systematic literature review consists of (i) database selection and bibliometric search; (ii) refining and sample selection; (iii) key concepts' extraction; and (iv) structuring the review based on the main topics extracted. Then, in-depth interviews were adopted to verify the results and to identify the challenges. The research procedure is illustrated in Figure 1.

### 2.1. Systematic Literature Review

Reviewing the existing literature is a fundamental aspect of any research field [14]. A systematic way of collecting and synthesizing previous studies could broadly be described as a literature review [15]. Besides, a solid base for more knowledge in a specific field by identifying the research status and defining theory development could be created through a thorough literature review [16]. Thus, reviewing the literature of previous studies in a systematic way is a critical characteristic of accelerating discipline development [15]. However, a systematic literature review methodology includes the strengths and weaknesses in comparison with other methods, such as expert reviews [17]. Considering the strengths, first, the location in which the research is carried out does not impact on the literature review, so it could be a free location. Second, unlike other research methods in which data are pertinent to a considerable research infrastructure such as academic and business networks, data for performing a systematic literature review are readily available. Third, scholars are able to refine the searches and analysis multiple times while doing a

literature review, but it is almost impossible to repeat the expert judgements overtime [18]. On the other hand, systematic reviews are time consuming when it comes to reading, understanding and completing the writing [19]. In addition, the review may not necessarily match the most updated field information due to the enormous number of written papers, as well as required time for searching, reading and writing. Moreover, a literature review cannot access all the published papers due to the limitations to access of databases [20]. For instance, in less common journals, there might be some relevant articles, which are not part of a well-known database such as Scopus. This study has benefited from a systematic keyword search in two broad databases of Scopus and Web of Science (WoS) to include all the possible publications in C&DWM in Australia to overcome these problems. The process of a systematic keyword search is discussed in the next section.

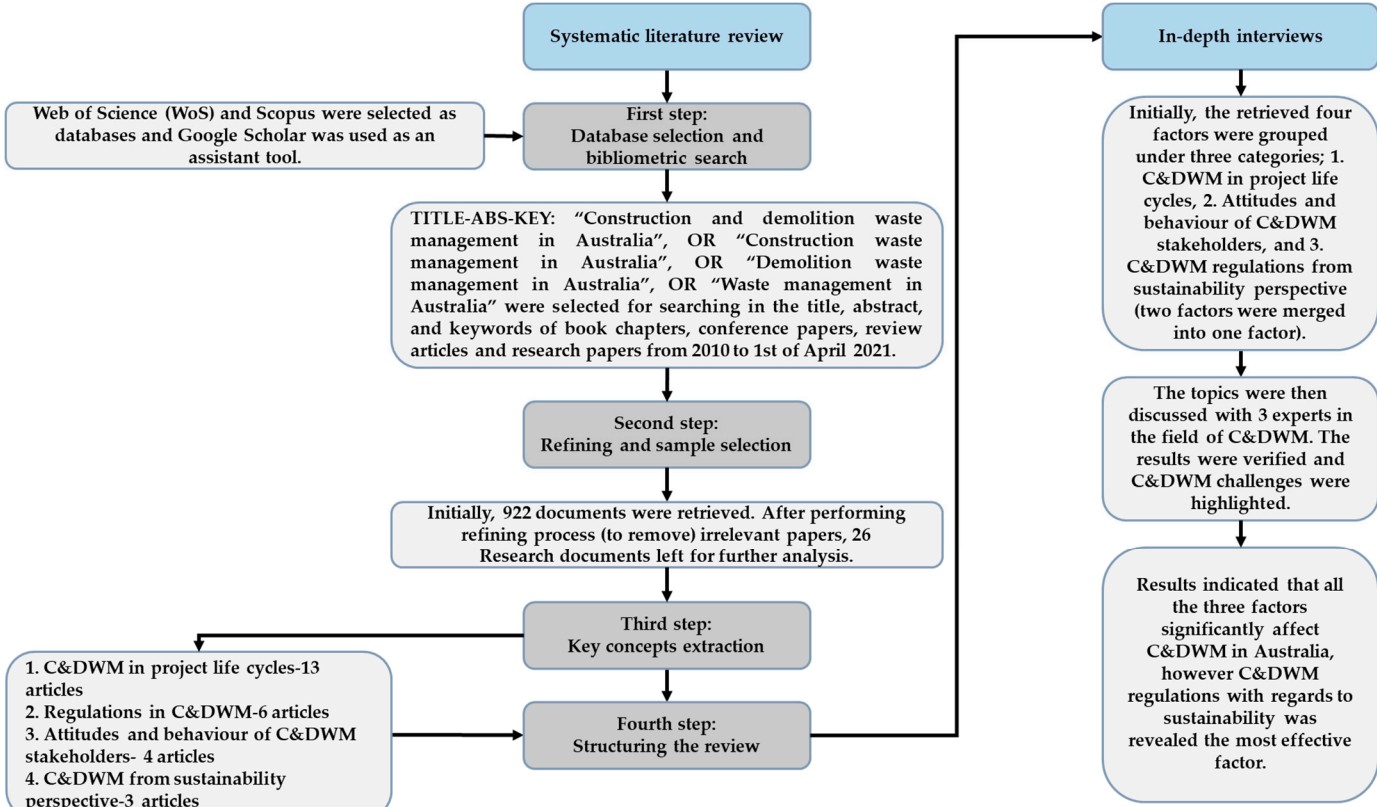

**Figure 1.** Research procedure.

### 2.1.1. Database Selection and Bibliometric Search
Database Selection

A number of databases are currently available for indexing scientific papers, such as Scopus, WoS, PubMed and Google Scholar. Some researchers have stated that the Scopus database outperforms WoS (e.g., [21]); however, other researchers have emphasized the comprehensiveness of WoS in comparison with other databases (e.g., [22]). Therefore, both Scopus and WoS databases were considered important for the extraction of C&DWM-related articles. In the meantime, by reviewing the journal list defined by previous reviews on C&DWM, it was found that almost all publications are listed in both databases. Therefore, Scopus and WoS were determined to be the primary sources for searching and filtering articles. Google Scholar was also used to update and add article samples as a search engine.

Bibliometric Search

TITLE-ABS-KEY: Four keywords of "Construction and demolition waste management in Australia", OR "Construction waste management in Australia", OR "Demolition waste

management in Australia", OR "Waste management in Australia" were selected for searching in the titles, abstracts and keywords of valid review and research articles, book chapters and conference papers from 2010 to 1 April 2021. After matching retrieved articles from both databases, a total of 922 articles were revealed.

### 2.1.2. Refining and Sample Selection

Articles identified from the initial sample selection were primarily scanned for relativity assessment to the research topic through reading their titles and abstracts. The aim of this phase was to extract the most relevant articles to C&DWM in Australia. In this step, the articles which addressed any other aspect of C&DW rather than management (e.g., C&DW optimisation), waste rather than C&DW (e.g., water waste, hazardous waste, industrial waste, municipal solid waste, etc) and articles pertinent to material waste rather than C&DWM (e.g., concrete waste), were discreetly removed from further analysis. In case there was any doubt in the relativity of articles to the research topic (C&DWM in Australia), the articles were thoroughly investigated by reading their whole content. Finally, 26 related articles were retrieved for further analysis.

### 2.1.3. Key Concepts Extraction

In this step, 26 retrieved articles are presented in Table 1, representing all the relevant studies to C&DWM in Australia. The key concepts of the articles are highlighted in column 2. The key concepts of the articles indicate the major researched topic of that article and form the foundation of structuring the review.

**Table 1.** Construction and demolition waste management research backgrounds in Australia (2010–2021).

| No | The Key Concept | Year | Source | Reference |
|----|----------------|------|--------|-----------|
| 1 | C&DWM regulations | 2021 | Sustainability (Switzerland) | [23] |
| 2 | C&DWM in project life cycles | 2021 | Australian Journal of Civil Engineering | [24] |
| 3 | C&DWM in project life cycles | 2020 | Construction Economics and Building | [25] |
| 4 | C&DWM in project life cycles/GRB tool for C&DWM | 2021 | Engineering, Construction and Architectural Management | [26] |
| 5 | C&DWM regulations | 2020 | Sustainability (Switzerland) | [27] |
| 6 | C&DWM regulations | 2020 | Resources, Conservation and Recycling | [3] |
| 7 | C&DWM in project life cycles/WDR | 2020 | Built Environment Project and Asset Management | [28] |
| 8 | C&DWM in project life cycles | 2020 | International Journal of Construction Management | [29] |
| 9 | C&DWM regulations | 2020 | International Journal of Environmental Technology and Management | [30] |
| 10 | C&DWM in project life cycles | 2018 | Journal of Green Building | [31] |
| 11 | Attitude and behaviour in C&DWM | 2018 | Sustainability (Switzerland) | [32] |
| 12 | C&DWM in project life cycles/C&DW quantification | 2018 | Facilities | [33] |
| 13 | Attitude and behaviour in C&DWM | 2018 | European Journal of Sustainable Development | [34] |
| 14 | C&DWM in project life cycles/Reusing C&DW | 2017 | International Journal of Construction Management | [35] |
| 15 | C&DWM in project life cycles | 2017 | PICMET 2016 | [36] |
| 16 | Sustainable C&DWM | 2017 | Procedia Engineering | [37] |
| 17 | C&DWM regulations | 2016 | Sustainability (Switzerland) | [38] |

**Table 1.** *Cont.*

| No | The Key Concept | Year | Source | Reference |
|----|-----------------|------|--------|-----------|
| 18 | C&DWM in project life cycles | 2015 | Resources, Conservation and Recycling | [39] |
| 19 | Attitude and behaviour in C&DWM | 2015 | International Journal of Construction Management | [40] |
| 20 | Sustainable C&DWM | 2015 | Book Chapter: Construction Safety and Waste Management: An Economic Analysis | [41] |
| 21 | C&DWM in project cycles | 2014 | Resources, Conservation and Recycling | [42] |
| 22 | C&DWM in project life cycles/C&DW recycling | 2014 | International Symposium on Automation and Robotics in Construction | [43] |
| 23 | C&DWM in project life cycles | 2014 | International Journal of Construction Management | [44] |
| 24 | C&DWM regulations | 2013 | Conference Proceedings | [45] |
| 25 | Sustainable C&DWM | 2013 | Journal of Legal Affairs and Dispute Resolution in Engineering and Construction | [46] |
| 26 | Attitude and behaviour in C&DWM | 2011 | CRIOCM 2011 | [47] |

Prior to structuring the review based on the key concepts of extracted articles, it is noteworthy to address a major bibliometric analysis pertinent to the extracted articles. Bibliometric analysis provides a wide range of information on published articles to the readers, such as the year of publication, authors' names and affiliations, keywords, funding resources, etc (e.g., [48–50]). However for small samples such as the current study, the major bibliometric information including the year of publication, author names, research institutes (affiliations) and keywords seems appropriate [51]. Since the year and source of publications have been addressed in Table 1, Figures 2 and 3 highlight the author names and affiliations (two major bibliometric information) with three or more articles, respectively. Meanwhile, Figure 4 represents the co-occurrence of author keywords in the retrieved articles in the VOSviewer version 1.6.13 [52]. The co-occurrence of keywords gives a clear picture from different aspects and directions of research topics to the researchers [5].

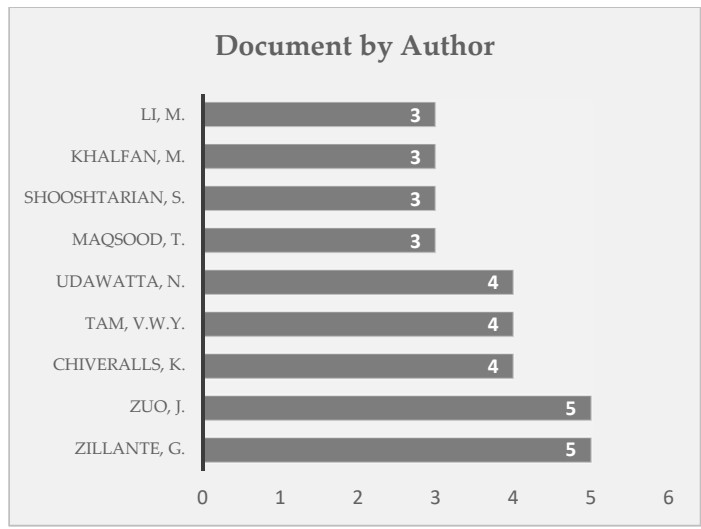

**Figure 2.** Documents by authors (authors with three or more articles).

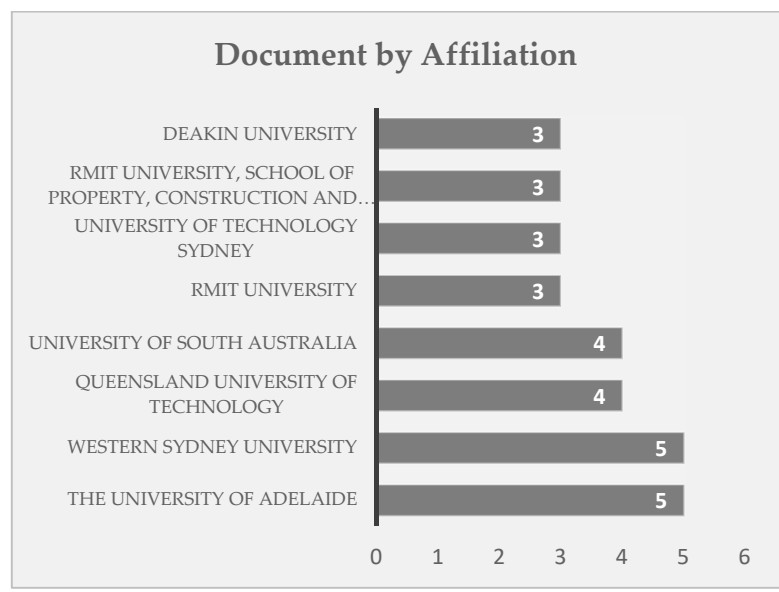

**Figure 3.** Documents by affiliation (affiliation with three or more articles).

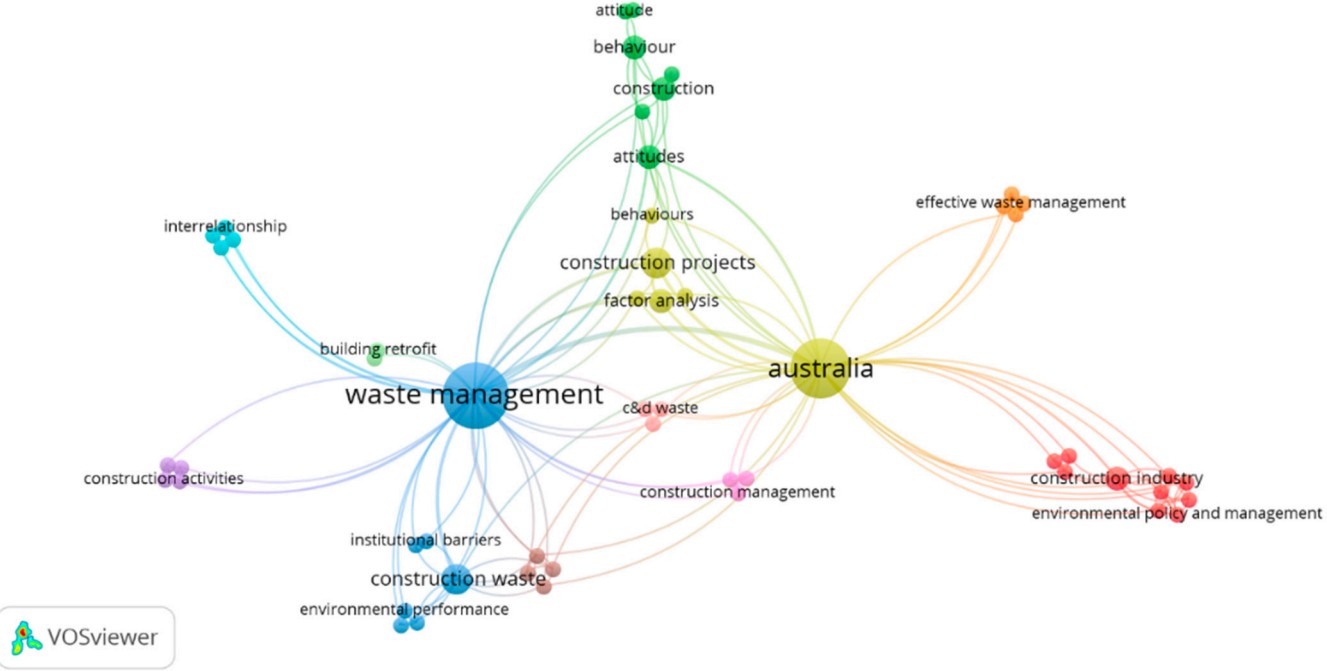

**Figure 4.** Co-occurrence of author keywords in C&DWM research in Australia.

As it can be seen in Figure 2, among the retrieved documents, Zillante, G. and Zuo, J. were the top authors in C&DWM research in Australia. Meanwhile, Figure 3 indicates that The University of Adelaide and Western Sydney University, with four articles each, were the top research institutes in addressing C&DWM in Australia. In addition to the bibliometric analysis, co-occurrence of author keywords has also been highlighted in Figure 4 via VOSviewer software. Co-occurrence of author keywords in a VOSviewer clarifies the keyword search directions in two ways; (i) it confirms whether the results from the bibliometric analysis matches the research objectives; and (ii) it is a useful tool for structuring the review, as it presents the pivotal keywords pertinent to the research topic [52,53]. "Waste management", "Australia", "construction projects" and "construction waste" were the most repeated keywords, with 12, 10, 3 and 3 repetitions, respectively. In

the next section, the key concepts of C&DWM research in Australia highlighted in Table 1 will structure the review.

### 2.1.4. Structuring the Review

From Table 1, it can be deduced that C&DWM research practices in Australia could be classified into four main categories including (i) C&DWM in project life cycles including C&DWM in construction sites, C&DW quantification, waste diversion rate assessment, green rating tools in C&DWM, and C&DW reduction, reuse and recycling (13 articles; 2–4, 7–8, 10, 12, 14–15, 18, and 21–23); (ii) regulations in C&DWM (6 articles; 1, 5–6, 9, 17, and 24; (iii) C&DWM with regards to the attitudes and behaviour of involved stakeholders (4 articles; 11, 13, 19, 26); and (iv) C&DWM from sustainability perspective (3 articles; 16, 20, 25). These four categories are discussed in Section 3.

### 2.2. In-Depth Interviews

In this research, online interviews were carried out to verify the results obtained from the systematic literature review and to identify the associated challenges. Interviews could be applied for results verification, validation [54,55] and to allow an in-depth analysis of problems [56,57]. Thus, construction professionals with C&DWM experience of ten years or more were targeted via the Google search engine and the LinkedIn platform within high-profile Australian construction companies [58]. They were approached by sending official invitation letters, as well as a research brief and questions. Finally, three experts (participants) were scheduled in an online interview. Therefore, the main findings from the systematic review were discussed with the experts via in-depth interviews. The experts included one civil and environmental engineer with extensive experience in environmental management and waste planning in consultant companies. The other was a construction manager with considerable experience in managing and directing construction projects in large contractor companies and the latter was a project/site engineer with a broad experience in supervising on-site C&D processes, as well as managing and directing sub-contractor (demolition) companies. The results obtained from in-depth interviews will be discussed in Section 4.

## 3. Results and Discussion

### 3.1. C&DWM in Project Life Cycles

This research classifies C&DWM in project cycles into five categories including C&DWM in construction sites, waste diversion rate assessment, C&DW quantification, green rating tools in C&DWM and studies with an emphasis on C&DW reduction, reuse and recycling; however, these studies had much in common.

These studies have mostly focused on C&DWM in construction sites. For instance, Li et al. [42,44] assessed the influence of the critical parameters of waste management in office building retrofit projects. These factors included the culture of the industry, existing building information, support and incentive from organisations, design and project delivery processes. It was also revealed that the organisational commitment, along with factors related to the construction life cycle including the plan, design and construction processes, are the critical factors affecting C&DWM in office retrofit projects. In another study, Brennan, Ding [43] compared Australia and Germany with respect to the waste reuse and recycling practices through a closed-loop system, despite the fact that C&DW is categorised slightly differently in the two countries. The results indicated that Germany has a higher recycling/recovery rate of C&DW, which can be attributed to the performance of pre-sorting and separating facilities. Similarly, the effective approaches to minimize C&DW in construction projects in Australia were investigated by Udawatta, Zuo [39] and five solution approaches were consequently presented for C&DWM including proper design and documentation, lifecycle management, strategic guidelines in waste management, innovation in waste management decisions, and team building and supervision. In addition, it was revealed that attitudinal approaches and C&DWM technologies such

as deficiencies associated with C&DWM should receive further attention in Australia. Besides, Rose, Manley [36] highlighted the inefficient reusing and recycling rate of C&DW in the Australian building projects and proposed a conceptual framework for on-site waste management. The core focus of the study was on waste stakeholders' attitude and integrated waste management processes from generation to disposal. Furthermore, Park and Tucker [35] investigated the reuse of construction waste and found that institutional barriers, including social, economic and political issues, are the biggest obstacles in reusing C&DW. Meanwhile, clients' lack of interest and demand, different attitudes towards reuse practices, and lack of training were revealed to be critical disincentives in reusing C&DW. Aligned with these studies, Udawatta, Zuo [31] investigated the factors that impede the Australian construction industry from optimum waste management and found that the rigidity of construction processes, the nature of waste management, experience, awareness and commitment, and characteristics of construction projects are the most critical factors.

Regarding C&DW quantification, Fini and Forsythe [25,33] investigated the amount and stream of generated waste in office buildings in Australia. It was revealed that almost half of the demolished waste in office fit-out goes to landfills in the form of mix waste. Moreover, furniture, fit-out assemblies and insufficient critical mass were three barriers in reusing and recycling C&DW in office projects. Furthermore, Newaz, Davis [29] studied C&DWM with regards to C&DW streams in Australia and found that a lack of unified regulations among states, the location of waste disposal points and people's attitudes towards C&DWM were the most important factors in C&DWM in Australia. The importance of C&DWM regulations in on-site C&DWM was also addressed by Doust, Battista [24], who developed front-end strategies by implementing initial material logistics management.

Considering the waste diversion rate, Ratnasabapathy, Alashwal [28] assessed the waste diversion rate (WDR) as a key indicator for effective waste management in Australia. The results revealed that mixed waste is the biggest waste stream in residential buildings. Considering the tools adopted in the Australian construction context to manage C&DW, Zillante, Chiveralls [26] investigated the green building rating systems in terms of waste management practices in educational buildings in Australia and found that waste management targets defined in green systems need more elaboration.

### 3.2. C&DWM Regulations

The second category of C&DWM in Australia was C&DWM regulations. From 26 retrieved research documents, 6 documents directly addressed C&DWM from a regulatory perspective. Some of these studies have focused on the comparison between C&DWM regulations in Australia and other countries. For instance, Li, Kühlen [45] compared Australia and Germany regarding C&DWM practices and regulations. It was found that the C&DW generation rate is faster than the recovery rate in Australia; however, the waste generation rate has been constant in Germany for many years, despite the fact that the recovery and recycling rates are increasing. The commitment to the regulations, benchmarking and supervision were the critical factors that should be improved in Australia. Similarly, Tam and Lu [38] performed a cross-jurisdictional assessment of C&DW generation and management in four regions comprising Europe, United Kingdom, Hong Kong and Australia. The results indicated that the C&DWM regulations in all regions has led to the decreasing rate of C&DW generation. However, Australia is far behind the recycling rate of C&DW among these countries. Table 2 highlights the major information pertinent to C&DWM in Australia, Germany, United Kingdom and Hong Kong.

**Table 2.** Construction and demolition waste management in Australia, Germany, United Kingdom and Hong Kong.

| Country | Construction and Demolition Waste Generation (Million Tonnes) | C&DW Recycling Rate | Waste Regulations | Refrences |
|---|---|---|---|---|
| Australia | 27 | 60% | National Waste Policy; Less waste, More resources (Department of Agriculture, Water and Environment), National Waste Policy; Action plan, and Jurisdictional regulations for C&DWM across Australian states and territories. | |
| Germany | 86 | 80–90% | European Laws (Waste Framework Directive (2008/98/EC), German Federal Law (1972), State law of Bundesländer, Municipal waste disposal law, and The Circualr Economy Act (KrWG) (2012) Key instruments: Selective demolition, C&DW sorting, separate collection, hazardous waste management and green public procurement. | |
| United Kingdom | 58 | 80–90% | Waste Framework Directive (2008/98/EC), Hazardous Waste Regulations, Landfill Legislation, European List of Wastes (Decision 2000/532/EC), Waste Producer's Responsibility, Specific legislation on C&DWM (e.g., site waste management plan in England), Landfill tax, and Restrictions/Ban on specific C&DW Key instruments: C&DW sorting, separate collection, hazardous waste management, green public procurement and landfill tax. | [11,59–62] |
| Hong Kong | 20 | 90% and above for inert waste | A major classification for inert/non-inert waste for C&DWM is consdidered, Waste Dispoal Ordinance (1980), Construction Waste Disposal Charging Scheme (2005), Waste management plan, Pilot recycling plant and Trip ticket system. | |

Other studies have investigated C&DWM regulations within Australia. For example, Shooshtarian et al. [27,30] studied the existing landfill levy across Australia and found that stakeholders with market incentive approaches are more numerous than those with a pecuniary impost approach. Similarly, Shooshtarian et al. [23] investigated the extended producer responsibility (EPR) scheme as a policy to prevent waste generation. It was found that there is a high support among various stakeholders for developing EPR and expanding the existing regulations. Furthermore, construction product lifecycle, time and cost, responsibility of manufacturers, multiplicity of stakeholders, complexity in EPR regulations usage and health and safety issues were identified as barriers to implementing EPR policy. Besides, Wu, Zuo [3] investigated the cross-regional management of C&DW with regards to the interstate C&DWM regulations. Their study indicated that the engendered C&DW in one state in Australia may be recycled in another state due to the lack of recycling facility. In addition, landfill levies, incentives for reusing and recycling C&DW, the market for recycled materials and the sustainability impact of C&DW were revealed to be critical.

### 3.3. Attitudes and Behaviour of C&DWM Stakeholders

Attitudes and behaviour of C&DW stakeholders play an important role in C&DWM. For instance, Tam and Shen [47] investigated the role of attitudes and habits in forming the

recycling habits of C&DWM stakeholders, showing the positive attitudes towards recycling habit, but not so strong behaviours. In addition, the work environment comprising the operating process and work routines are not adequately prepared to be aligned with the recycling behaviour and affect the recycling outcomes. Similarly, Udawatta, Zuo [40] studied the attitudes and behaviours of stakeholders towards waste management, resulting in majority of decisions being on the basis of the financial returns in construction projects. In addition, private developers were revealed to be more price-driven in comparison with government clients, and contractors were in favour of financial incentives. In another study, Forghani, Sher [34] investigated the role of attitudes of demolition contractors on the management of their operations and how these attitudes could potentially motivate the reuse of building parts. The results indicated that over one-third of demolition contractors are without the guideline, strategy or goal to reuse the building components. Furthermore, Tam, Le [32] studied the recycling attitudes and behaviour of the C&DWM stakeholders and found that remedies such as waste management method improvement, work statements' provisions, legislation and market-driven developments, sharing research and applications in sub-industries and developing communication are the prominent factors.

### 3.4. C&DWM from a Sustainability Perspective

C&DWM from a sustainability perspective is mostly referred to as regulations to manage C&DW, unless it measures life cycle analysis. For instance, Tam Vivian and Zeng [46] developed sustainability function indicators to verify the sustainable performance surpass of local residential buildings with regards to C&DWM and it was revealed that legislation for the enforcement of sustainability in construction is critical for waste management. In another study performed by Li and Du [41], it was revealed that the government of Australia has a great impact on contractors to utilize prefabrication techniques, reused and recycled materials, a waste minimization plan, and finally to increase the recovery rate in order to reduce C&DW. Furthermore, Crawford, Mathur [37] investigated the factors that influence the environmental performance of C&DW in remote communities in Australia. The factors included on-site waste management cost and time, culture of industry, lack of education among personnel, lack of financial incentives and having a preference about project priorities rather than waste management. By assessing C&DWM from a sustainability perspective, it could be deduced that these practices include the environmental, social and economic aspects of C&DW and are mostly linked to the regulations pertinent to C&DWM.

Discussing the results followed by the review revealed that the four categories of factors pertinent to C&DWM in Australia, including C&DWM in project life cycles, C&DWM regulations, attitudes and behaviour of C&DWM stakeholders and C&DWM from a sustainability perspective, could be rearranged and grouped into three categories. Since C&DWM regulations and C&DWM from sustainability are closely correlated in the Australian context, these factors are merged and form a new category, C&DWM regulations with regards to sustainability. Thus, the three categories that affect C&DWM in Australia are (i) C&DWM in project life cycles; (ii) attitudes and behaviour of C&DWM stakeholders; and (iii) C&DWM regulations with regards to sustainability. This classification is also consistent with the global categorization of contributing factors to C&DWM [11]. Table 3 summarizes these factors and their components.

**Table 3.** Factors contributing to construction and demolition waste management in Australia.

| Paper Numbers in Table 1 | Main Categories | Major Components | References |
|---|---|---|---|
| 2–4, 7, 8, 10, 12, 14, 15, 18, and 21–23 (13 articles) | 1. C&DWM in project life cycles | 1. The culture of construction industry, the role of incentive and organisational support, as-built information of buildings, design and the process of delivery of the project (plan, design, and construction) are factors affecting C&DWM in office retrofit projects. 2. Pre-sorting and separating facilities have a great impact on reusing and recycling of C&DW. 3. Five solutions for waste management in construction projects were supervision, having guidelines for waste management, accurate design and documentation, innovative decisions and life cycle management of waste. 4. The deficiency in reusing/recycling of C&DW in Australia can be attributed to the technological and attitudinal factors. 5. Social, economic and political issues, as well as attitudinal approaches, are the biggest obstacles in reusing C&DW in Australia. 6. The rigid nature of the construction industry, specific characteristics of construction projects, commitment, experience, and awareness and the embryonic nature of waste management are factors that impede the Australian construction industry from maximum waste management. 7. Mix-waste is a big issue to deal with in office renovation, as well as in residential demolition projects. 8. A lack of unified regulations for C&DWM across Australian states, inappropriate attitude of stakeholders towards waste management and the limited number of recycling facilities are important factors in waste management in Australia. | [24–26,28,29,31,33,35,36,39,42–44] |
| 1, 5, 6, 9, 17, 24, 16, 20, and 25 (9 articles) | 2. C&DWM regulations with regards to sustainability | 1. C&DWM regulations, benchmarking and supervision should be improved in Australia. 2. Landfill levy, incentive/punishment mechanisms, adequate number of recycling facilities, market for recycled products and sustainability impact of C&DW are effective factors in C&DWM in Australia. 3. Legislation should mandate sustainable practices of C&DWM. 4. Time and cost associated with waste management, lack of education and common perception towards waste management, unclear guidelines of waste management and having a preference for project priorities rather than waste management are other important factors in C&DWM in Australia. | [3,23,27,30,37,38,41,45,46] |
| 11, 13, 19, and 26 (4 articles) | 3. Attitudes and behaviour of C&DWM stakeholders | 1. Attitudes and behaviour of stakeholders involved in C&DWM should be in the same direction for optimum waste management. 2. Financial return plays a crucial role in altering the attitude and behaviour of C&DWM stakeholders. 3. Training and communication are important factors to improve stakeholders' attitudes towards better waste management. | [32,34,40,47] |

In the next section, these three categories will be verified through in-depth interviews.

## 4. Results Verification via In-Depth Interviews

The general procedure of in-depth interviews was primarily discussed in Section 2.2. The interview consisted of general and specific questions in four main sections. The first section focused on the potential gaps of C&DWM in Australia. The rest were about C&DWM in project life cycles, attitudes and behaviour of C&DWM stakeholders and C&DWM regulations with regards to sustainability. Initially, the three interviewees agreed that all the three identified categories including C&DWM in project cycles, attitudes and behaviour of C&DWM stakeholders, and C&DWM regulations with regards to sustainability are critical components of C&DWM in Australia.

Considering the potential gaps of C&DWM in Australia, the respondents stated that the lack of relevant data to C&DW (C&DW as an independent waste classification) amount and stream is one major obstacle in C&DWM in Australia, as data are more related to waste in its general form rather than C&DW. Besides, data pertinent to C&DW in Australia are more collected at recycling facilities and are more related to recycled/recovered waste; however, based on interviewees' opinions, equal consideration should be given to C&DW data transparency from project initiation to closure with more emphasis on collecting data pertinent to generated C&DW at construction sites. This data inefficiency has also been highlighted in the Waste National Report 2020 [6]. Moreover, the respondents pointed to the dispersion of C&DWM instructions, which are addressed in several publications such as reports, standards, toolkits and strategies as another obstacle of C&DWM in Australia, since there is not a unified reference for C&DWM on a national scale. Furthermore, the respondents unanimously recommended that waste sorting and separation at source should be the primary focus of C&DWM in Australia. In addition, the respondents pointed at an inadequate use of digital tools and technologies (e.g., prefabrication) to reduce the amount of C&DW in construction projects.

Regarding C&DWM in project life cycles, two interviewees pointed to the planning and design phase in the initiation of any construction project, in which extra attention should be paid to the selection of reusable and recyclable materials rather than imposing C&DWM to the construction/demolition stage. Furthermore, one respondent addressed material ordering, handling and storage as important factors during the procurement phase of construction projects. Meanwhile, the interviewees believed that C&DWM during construction/demolition phases should be the first priority of C&DWM in project life cycles, which could be achieved through an accurate supervision and waste management plan. These findings are also consistent with the related literature [31,39,42].

Considering the attitudes and behaviour of C&DWM stakeholders, the interviewees agreed that even though the attitude of C&DWM stakeholders is a pivotal factor in C&DWM in Australia, it does not necessarily motivate or force C&DWM stakeholders to adopt and implement regulations pertinent to C&DWM (regulations are more important than attitudes). Besides, two interviewees stated that generally, clients/developers have the greatest impact on C&DWM by their attitudes and behaviour. The related literature also indicates that clients/developers have a decisive role in selecting and auditing other stakeholders with respect to C&DWM [40]. Thus, it is essential to alter clients' attitudes towards waste management from a profit-based approach to sustainability considerations. Moreover, two interviewees criticized the practices of private developers such as selecting the lowest bidders for construction/demolition projects without much concern about their waste management performances. Aligned with [39,40], the respondents confirmed that the clients have prequalification processes to choose potential contractors/demolishers in government projects, which may, in the case of private clients, be based on cronyism.

With respect to C&DWM regulations with regards to sustainability, the three interviewees agreed that the governmental regulations in C&DWM (i.e., landfill levies) is a significant factor affecting C&DWM in Australia. However, more supervision on demolition companies should be performed to prevent them from illegal dumping (e.g.,

dumping asbestos illegally). In addition, financial incentives in promoting C&DW reuse and recycling and developing a market for recycled material should be developed by the Australian government. Moreover, the interviewees agreed that the economic factor is the most significant influencing factor in pursuing C&DWM from a sustainability perspective, even higher than environmental/social considerations. It was therefore suggested that the government must either provide attractive incentives versus rigorous punishment mechanisms for better C&DWM; however, the role of supervision of the mechanisms is also important. One interviewee stated that these incentive/punishment mechanisms should be addressed in contract documents. Another interviewee with experience in managing construction projects in the different states and territories of Australia pointed to the lack of unified regulations in different states and territories, which could also lead to inefficient management of C&DW, especially when C&DW is managed cross-regionally. These findings were also consistent with the relevant literature [3,27]. Respondents were also asked to rank the C&DWM categories, and they agreed that C&DWM regulations with regards to sustainability is the most important factor of C&DWM in Australia. In addition, the attitudes and behaviour of C&DWM stakeholders and C&DWM in project life cycles were other important factors of C&DWM in Australia. The main findings of interviews are represented in Table 4.

**Table 4.** Main findings of interviews with three C&DWM experts.

| Interviewee | Construction and Demolition Waste Management Gaps | (A) C&DWM in Project Life Cycles * | (B) Attitudes and Behaviour of C&DWM Stakeholders ** | (C) C&DWM Regulations with Regards to Sustainability *** | C&DWM in Australia |
|---|---|---|---|---|---|
| Interviewee 1 | 1. Insufficient/inefficient data pertinent to C&DW, lack of unified reference for C&DWM in a national scale, and dispersed instructions, standards, reports, etc. for C&DWM across state and territories governments. 2. Inadequate utilisation of tools and technologies in managing C&DW. | First priority: C&DWM in construction/demolition stages, Second priority: C&DWM in planning/design stages. | Clients and developers have great impacts on C&DWM in Australia. Contractors and consultants also affect C&DWM in Australia. | All interviewees stated that economic factor is among the most important factors in the structure of C&DWM regulations. Landfill levies, incentive/punishment mechanisms for C&DWM, profit from selling recycled material are important factors in regulating and managing C&DW in Australia. | Priorities: Factors C, B, and A, respectively. |
| Interviewee 2 | | First priority: C&DWM in construction/demolition stages, Second priority: C&DWM in procurement phase. | All stakeholders including contractors, consultants, clients, etc. have equal impacts on C&DWM in Australia. | | Factor C is the most effective factor in C&DWM in Australia; and factors B and A have equal importance. |
| Interviewee 3 | | First priority: C&DWM in construction/demolition stages, Second priority: C&DWM in planning/design stages. | Clients and developers have great impacts on C&DWM in Australia. This is followed by contractors. | | Priorities: Factors C, A, and B, respectively. |

* Construction and demolition waste management in project life cycles; ** Attitudes and behaviour of construction and demolition waste management stakeholders; *** Construction and demolition waste management regulations with regards to sustainability.

## 5. Conclusions

This study undertook a review on C&DWM in Australia. Primarily, a structured review with the support of a keyword searching approach was performed to identify the most relevant studies to C&DWM. In this step, 26 articles relevant to C&DWM in Australia were retrieved. These articles were primarily classified into four categories of C&DWM in project life cycles, regulations in C&DWM, C&DWM with regards to the attitudes and behaviour of involved stakeholders and C&DWM from a sustainability perspective. Then, after performing an in-depth review of articles, it was found that these factors could be converted into three factors by merging C&DWM from a sustainability perspective to C&DWM regulations. Therefore, finally, three sub-factors of C&DWM in project life cycles,

attitudes and behaviour of C&DWM stakeholders, and C&DWM regulation with regards to sustainability were left for further analysis. Next, these factors were validated through three in-depth interviews with three C&DWM professionals. The results were consistent with the relevant literature. Finally, it was revealed that C&DWM regulations with regards to sustainability were the most important factor in C&DWM.

Moreover, from interviews with professionals, it was revealed that data pertinent to C&DWM should be highlighted in all phases of projects from initiation to closure. In addition, more attention should be given to C&DWM at a national scale and fragmented reports, standards, regulations, etc.; in the state and territories, the scale should be unified in order to be more effective. Besides, results indicated that less attention was given to the role of tools and technologies in C&DWM in Australia. For instance, there was only one study that referred to the green building rating tool in C&DWM [26].

A structured review of C&DWM studies, as well as identifying and classifying the factors that affect C&DWM in Australia, can provide useful insights for both academics and professionals in C&DWM. Industry professionals could benefit from the current study, as it gives a clear picture of factors that affect C&DWM. Thus, by implementing strategies that address these factors, C&DW can be better managed within their organisations. Academics can also benefit from this research by underpinning their future research topics on the retrieved factors pertinent to C&DWM.

The current study addressed C&DWM regulations from a sustainability perspective as a factor of C&DWM in Australia. However, in-depth analysis of these regulations, as well as accurate comparisons of C&DWM regulations in Australia with other countries, can be considered as a limitation of this study and a good topic for future research. In addition, future studies can empirically investigate the factors that influence C&DWM in Australia.

**Funding:** This research received no external funding.

**Conflicts of Interest:** The authors declare no conflict of interest.

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
