# Peer review of "A Systematic Review of Construction and Demolition Waste Management in Australia: Current Practices and Challenges"

_recycling, doi:10.3390/recycling6020034_

Round 1

Reviewer 1 Report

Section 2.1  can be summarize in one paragraph

Section 3.2. A comparison table containing the main common point and the main differences among regulations of different countries would be helpful and will give the paper more relevance

Section 4. The main results pointed out by the three experts could be presented in a matrix in order to rank the most relevant and easily take a look at the priorities to be solved.

Author Response

Reviewer 1:

Section 2.1  can be summarize in one paragraph.

Authors’ response: Many thanks for your precise consideration. Please see section 2.1 in the revised manuscript. The strengths and weaknesses of systematic literature review were summarised and drafted along with each other.

Section 3.2. A comparison table containing the main common point and the main differences among regulations of different countries would be helpful and will give the paper more relevance.

Authors’ response: Many thanks for your precise consideration. The major information pertinent to CDWM (a comparison between Australia and countries mentioned in the text from statistical and regulatory framework perspective) have been highlighted in Table 2. Please see section 3.2. Please note that an in-depth comparison on differences of CDWM regulations in different countries can be defined as a research topic itself. The authors also addressed this point as the limitation of the study in the Conclusion section.

Section 4. The main results pointed out by the three experts could be presented in a matrix in order to rank the most relevant and easily take a look at the priorities to be solved.

Authors’ response: Many thanks for your precise consideration. Table 4 in Section 4 represents the main findings.

Reviewer 2 Report

This is an interesting article that gives an outlook about the general state of art of CDWM in Australia from research (articles) and operational (interviews) points of view.

I appreciated how this article was friendly readable (good english, some typos found).

I would appreciate its acceptance within the recycling journal.

Furthermore, I believe that these suggestions would enhance the quality of this paper. Some of them are requested (about expert interviews) but others may be optional and may be the subject of perspective study.
- About expert interviews: It would be highly appreciable to include the questionary used for the interviews so the reader can reflect/link the state of art outputs with the interview results. A proposition from the expert based papers would be to provide the blank questionary used for interview.
- this paper resumes the challenges of CDWM in Australia. In some sections, comparison was made with other countries (germany...), which was highly valuable for the reader as well as for validating your statements. Another further comparison could be made in order to check if the factors determined are the same in other countries or is it a specific case of Australia ?

  • can you reformulate line 226 (maybe you mean ...many in common)
  • line 79: two research procedures
  • line 79: the To ?

- I also think that for a better spread and referencing of this paper, C&DW should be used as a keyword, rather than CDW...at least it is the consensus in EU

Author Response

Reviewer 2: This is an interesting article that gives an outlook about the general state of the art of CDWM in Australia from research (articles) and operations (interviews) points of view. I appreciated how this article was friendly readable (good English, some typos found). I would appreciate its acceptance within the recycling journal.

Furthermore, I believe that these suggestions would enhance the quality of this paper. Some of them are requested (about expert interviews) but others maybe optional and maybe subject of perspective study.

- About expert interviews: It would be highly appreciable to include the questionary used for interviews so the reader can reflect/link the state of art outputs with the interview results. A proposition from the expert based papers would be to provide the blank questionary used for interview.

Authors’ response: Many thanks for your precise consideration. As this paper is part of my PhD research project and I have not yet defended and submitted my thesis for examination, my supervisors have kindly requested that the sharing of questionnaire and interview questions would be exempted because as a part of approved ethics application, we can share these documents once the PhD thesis is published. Your help would be much appreciated.

- this paper resumes the challenges of CDWM in Australia. In some sections, comparison was made with other countries (Germany…), which was highly valuable for the readers as well as for validating your statements. Another further comparison could be made in order to check if the factors determined are the same in other countries or is it a specific case of Australia?

Authors’ response: Many thanks for your precise consideration. Considering CDWM regulations in Australia and other countries, a table (Table 2) was added to section 3.2 with major information pertinent to CDWM in Australia as well as Germany, United Kingdom, and Hong Kong reflecting the comparisons made in study of Tam and Lu (2016).

Meanwhile,  if the classification of factors that contribute to CDWM in Australia is the case, a statement was added to section 3.4, lines 351 and 352 confirming that this classification also exists in a global perspective.

- Can you reformulate line 226 (maybe you mean… many in common)

Authors’ response: Thanks for your precise consideration. The sentence was revised. “… had many in common.”

- Line 79: two research procedures

Authors’ response: Thanks for your precise consideration. The sentence was revised. “Two research procedures.”

- Line 79: the To?

Authors’ response: Thanks for your precise comment. “The” was removed from the sentence.

- I also think that for a better spread and referencing of this paper, C&DW should be used as a keyword rather than CDW …at least it is the consensus in EU.

Authors’ response: Many thanks for your precise consideration. CDW and CDWM were replaced by C&DW and C&DWM, respectively.

----------------------------------------------------------------------------------------

Round 2

Reviewer 1 Report

I think the paper has been improved